# Effects of Using Sitting Position versus Lithotomy Position during the Second Stage of Labour on Maternal and Neonatal Outcomes and the Childbirth Experience of Chinese Women: A Prospective Cohort Study

**DOI:** 10.3390/healthcare11222996

**Published:** 2023-11-20

**Authors:** Li Fu, Jing Huang, Danxiao Li, Huide Wang, Lili Xing, Tao Wei, Rui Hou, Hong Lu

**Affiliations:** 1School of Nursing, Peking University, Beijing 100191, China; fuli@bjmu.edu.cn; 2Division of Care for Long Term Conditions, Florence Nightingale Faculty of Nursing, Midwifery and Palliative Care, King’s College London, London SE1 8WA, UK; jing.huang@kcl.ac.uk; 3Department of Obstetrics and Gynaecology, Peking University People’s Hospital, Beijing 100044, China; lixiaodan6390@163.com (D.L.); 13691379900@139.com (L.X.); 4Department of Obstetrics, Beijing Hospital, Beijing 100730, China; wangdehui1997@163.com (H.W.); weitao2294@bjhmoh.cn (T.W.)

**Keywords:** sitting position, labour stage, second, term birth, maternal and neonatal outcomes, childbirth experience, prospective studies

## Abstract

Existing research concerning the effects of the sitting birth position during the second stage of labour on maternal and neonatal outcomes remains controversial, and there is a lack of studies to explore its effect on the childbirth experience. The objective of this study is to explore whether the sitting birth position would influence maternal and neonatal outcomes, as well as the childbirth experience. The prospective cohort design was conducted in the study from February to June 2023, a total of 222 women (including primiparous women and multiparous women) were enrolled in our study, and they were divided into the sitting position cohort (n = 106) or the lithotomy position cohort (n = 116). The pre-designed questionnaire and Childbirth Experience Questionnaire (CEQ) were used for data collection during hospitalisation. Chi-square, Fisher’s exact test, *t*-tests, or the Mann–Whitney U test were utilised to assess differences between groups. Multivariate linear regression and logistic regression were employed to control possible confounders. The study found that primiparous women in the sitting position cohort had a shorter duration of the second stage of labour, higher spontaneous vaginal birth rates, lower episiotomy rates, and a better childbirth experience (*p* < 0.01). After adjusting for confounding factors through multiple linear and logistic regression analyses, the results remained consistent with those reported above. No neonate in each cohort had Apgar scores at 1 min and 5 min postpartum less than 7 or a Cord artery pH less than 7.00, regardless of parity. Based on the findings, we recommend that women could take the sitting birth position into account when giving birth for a positive childbirth experience, especially for primiparous women. The study could also serve as a reference for healthcare providers in the management of childbirth positions and the development of high-quality maternal care.

## 1. Introduction

With the rapid advancements in maternal and neonatal healthcare, the primary focus has shifted from merely reducing maternal and neonatal mortality to improving the quality of midwifery services and optimising childbirth management, so as to promote natural birth and positive childbirth experience [1]. The second stage of labour is an important component of natural labour and plays a vital role in ensuring the health of women and newborns [2]. Furthermore, the duration of the second stage of labour usually serves as a core element of the management in this period [3]. According to The American College of Obstetricians and Gynecologists (ACOG) Practice Bulletin, the mean duration of the second stage of labour is 54 min for primiparous women and 19 min for multiparous women, with an additional 25 min for women with epidural analgesia [4].

A “prolonged second stage” is associated with multiple negative maternal outcomes, such as instrumental assisted birth and conversion to caesarean section [3]. In addition, it can also result in severe perineal lacerations, haemorrhage, and intrauterine foetal distress [5].

Improved scientific management of maternal positions in the second stage of labour has the potential to influence the duration of the second stage of labour and other maternal and neonatal outcomes [6]. As recommended by the World Health Organization (WHO), women could choose the appropriate upright birth positions of their own free will during the second stage of labour for a positive childbirth experience [7]. The sitting position, as an important upright position [8], has gained more attention in recent research studies exploring its impact on maternal and neonatal outcomes [9,10].

The findings of existing studies exploring the effects on the maternal and neonatal outcomes of the sitting position in the second stage of labour remain inconsistent [11,12]. Regarding the duration of the second stage of labour, some studies reported a significant shortening in the sitting position compared with the lithotomy position [13,14]. However, other studies showed adverse results for the effects of the sitting position on the duration of the second stage of labour [15,16]. In addition, there are mixed results regarding the effects of the sitting position compared to the lithotomy position on other maternal and neonatal outcomes, such as the birth mode, perineal lacerations, blood loss, or Apgar scores [17,18,19]. Therefore, it appears that current research regarding the influence of the sitting position during the second stage of labour on maternal and neonatal outcomes remains controversial, underscoring the need for further high-quality studies to investigate this topic.

Childbirth experience is considered an important aspect of ensuring high-quality maternal care based on the recent WHO recommendations, which could reflect women’s expectations and feelings throughout the labour process [7]. Ganapathy et al. [12] used a self-designed questionnaire to assess maternal birthing experiences between sitting and lithotomy position groups, showing that a higher number of women in the sitting position group reported feeling comfortable and had a positive perception of participation. However, another study [20] found no significant differences in any aspect of the mother’s experience (feeling of birth, pain, anxiety, and fatigue) for different positions. Currently, there is an insufficient number of studies comparing the sitting position to the lithotomy position during the second stage of labour in terms of childbirth experience assessed using valid scales. More research in this area is needed in the future.

Thus, the aim of our study was to explore the effects of using the sitting position versus the lithotomy position during the second stage of labour on maternal and neonatal outcomes, as well as women’s childbirth experience. The findings from this study could shed light on birth position management in clinical practice and help improve the quality of maternal care.

## 2. Materials and Methods

### 2.1. Study Design and Participants

This was a prospective cohort study conducted from February to June 2023 at Peking University People’s Hospital, Beijing, China. Women were eligible if they (1) intended to have a spontaneous vaginal birth; (2) were aged between 20–35 years old; (3) had a gestational age range from 37 + 0 to 41 + 6 weeks; (4) had a singleton cephalic presentation; (5) could communicate normally and participate voluntarily. They were excluded if they had (1) an abnormal foetal position (e.g., persistent occipital-transverse and occipital-posterior position, etc.); (2) severe pregnancy or childbirth complications, such as severe eclampsia, heart disease, cephalic presentation dystocia, etc.; (3) pelvic stenosis; (4) precipitate labour. The eligible women were informed of the details of our study by the attending midwife and decided whether or not to participate in the study. Finally, the participants were divided into the sitting position cohort and the lithotomy position cohort of their own free will. This study obtained ethical approval from the institutional review board (2022PHB188-001) and obtained written consent from all participants. The Strengthening the Reporting of Observational Studies in Epidemiology (STROBE) statement guidelines for reporting observational studies were adopted to report our study [21].

### 2.2. Study Variables

Demographic and clinical baseline data included parity, maternal age, education, gestation weeks, caesarean section history, pre-pregnancy BMI, weight gain, position of foetus, oxytocin use, labour analgesia, baby’s birth weight and length, position in the first stage of labour, and low-risk complications (such as premature rupture of membrane, mild pre-eclampsia, pregnancy with diabetes, pregnancy with mild arrhythmia, and pregnancy complicated with hysteromyoma). The primary outcome of this study was the duration of the second stage of labour. Secondary outcomes included the birth mode (including spontaneous vaginal birth, instrumental assisted birth, and caesarean section), episiotomy, perineal injuries including complete and laceration (degree of severity was determined according to ACOG guidelines [22]), postpartum 2h-haemorrhage (>500 mL), newborn Apgar score at one, five and ten minutes postpartum, artery pH, and the childbirth experience. Referring to previous studies [10,23,24], the identified possible confounders included gestation weeks, age, pre-pregnancy BMI, weight gain, baby’s birth weight, oxytocin use, epidural analgesia, and low-risk complications, which might impact the effects of the maternal positions on maternal and neonatal outcomes.

### 2.3. Data Collection

The pre-designed questionnaire was used for data collection, including maternal demographic and clinical baseline data and study outcomes. The information regarding maternal demographic and clinical baseline data was prospectively collected by reviewing hospital records in the first stage of labour. Data for study outcomes were similarly gathered from hospital records after childbirth by the researcher. In addition, women were asked to complete the Chinese version of the Childbirth Experience Questionnaire (CEQ) within 24 h of the birth to obtain their childbirth experience [25]. If women had any questions about the Childbirth Experience Questionnaire (CEQ), the midwife who did not participate in the childbirth process would explain it to them and collect the completed questionnaire. The Childbirth Experience Questionnaire (CEQ) [26] was developed in Swedish by Dr. Dencker et al. It has been widely used as a reliable instrument to assess women’s perceptions and experiences during childbirth. The adapted Chinese version of the CEQ contained four dimensions with 19 items to evaluate women’s professional support, self-ability, self-perception, and sense of participation. High scores demonstrate better childbirth experiences.

### 2.4. Statistical Analysis

The primary outcome-duration of the second stage of labour was selected to conduct sample size calculations [27]. According to a similar study [28], the standard deviation (SD) of the duration of the second stage of labour was defined as 22 min, and the mean value was 26.36 min for the exposed group, while it was 35.03 min for the control group. We assumed an alpha at 0.05, a power of 90%, and two-tailed tests. Eventually, a total of 216 participants were needed. SPSS version 27.0 software [29] was used for statistical analysis. Demographic and clinical baseline data were summarised with descriptive statistics. Count data, such as the rates of perineal laceration and episiotomy, etc., were analysed through the chi-square or Fisher’s exact tests for comparisons between groups. For continuous variables, such as the duration of the second stage of labour, the test of normality was used to check the normality of variables, and the student’s *t*-test was utilised to compare the differences between groups if the data followed the normal distribution. The Mann–Whitney U test was conducted if the data did not conform to the normal distribution. Multivariate linear regression and logistic regression were employed to control possible confounders [10,23,24] when comparing the differences in primary and secondary outcomes between the two cohorts. The sample of the multiparous women was limited to the multivariate regression analysis of the maternal and neonatal outcomes. All comparative analyses were recognised as statistically significant if the *p*-value was less than 0.05.

## 3. Results

### 3.1. Demographic and Clinical Characteristics of Participants

A total of 347 women were approached, and of these, 125 women were excluded for not fitting the inclusion criteria or refusing to participate. As a result, 222 women (183 primiparous women, 39 multiparous women) participated in the study, 106 in the sitting position cohort and 116 in the lithotomy position cohort. Among primiparous women, there were 91 women in the sitting position cohort and 92 women in the lithotomy position cohort. For multiparous women, 15 were in the sitting position cohort and 24 were in the lithotomy position cohort. The flowchart of enrolment is shown in Figure 1. Of these 222 participants, the mean age was 30.97 (SD = 2.66) years, and the mean gestation was 39.69 (SD = 0.97) weeks. Most participants had a bachelor’s degree or higher and had no caesarean section history. The mean maternal weight gain during pregnancy was 14.02 (SD = 7.58) kg, and all the foetal positions were occiput anterior. Detailed demographic characteristics and birth information by parity and position are shown in Table 1. The demographic and clinical baseline data were generally similar between the sitting- and supine-birth cohorts, regardless of parity.

### 3.2. Comparison of the Primary and Secondary Outcomes of Childbirth between Cohorts

Table 2 presents the comparison results of maternal and neonatal outcomes between the two cohorts. Among primiparous women, the duration of the second stage of labour in the sitting position cohort was significantly shorter than that in the lithotomy position cohort (*p* < 0.001). There was no significant difference in the duration of the first stage of labour between the two cohorts (*p* = 0.455). A higher rate of spontaneous vaginal birth (*p* = 0.001) was observed among women in the sitting-birth cohort (93.4%, 85/91) than women in the lithotomy-birth cohort (75%, 69/92). For episiotomy, a significantly lower rate was found in the sitting-birth cohort compared to the lithotomy-birth cohort (*p* = 0.001). Perineal injuries among non-episiotomy samples (in terms of complete and laceration) (*p* = 0.725) and postpartum 2h-haemorrhage (*p* = 0.654) were not significantly different in the two cohorts. Among multiparous women, no statistical difference was found in all maternal outcomes between cohorts. None of the infants had an Apgar score of less than 7 at 1 min, 5 min, or 10 min after birth, and the cord artery pH of all infants was higher than 7.0. Therefore, there were no cases of neonatal asphyxia in either the sitting position cohort or the lithotomy position cohort.

As shown in Table 3, among primiparous women, the sitting position cohort reported significantly higher scores on the CEQ questionnaire (*p* < 0.001) across all four dimensions, including professional support (*p* = 0.000), self-ability (*p* = 0.000), self-perception (*p* = 0.000), and sense of participation (*p* = 0.000) compared to the lithotomy position cohort. For multiparous women, the CEQ scores did not differ significantly between the two cohorts (*p* = 0.074), except for the dimension of self-support (*p* = 0.019). Using the multivariate linear regression and the logistic regression analysis for primiparous women to adjust the potential impact of the confounders, including gestation weeks, age, BMI, weight gain, baby’s birth weight, oxytocin uses, epidural analgesia, and low-risk complications, we gained the same results as above. We found that the sitting position has an independent impact in terms of the duration of the second stage of labour, birth mode, episiotomy, and CEQ scores. The sitting-birth cohort showed a shorter duration of the second stage of labour (*p* < 0.01), more positive childbirth experience (*p* < 0.01), and higher rates of spontaneous vaginal birth (*p* < 0.01) and episiotomy (*p* < 0.01). However, there was no significant difference in perineal injuries (*p* > 0.05) or postpartum 2h-haemorrhage (*p* > 0.05). The details are shown in Table 4.

## 4. Discussion

This paper was designed to explore the effects of using the sitting position versus the lithotomy position during the second stage of labour on maternal and neonatal outcomes, as well as women’s childbirth experience, which aimed to provide a reference for the practice of childbirth positions and improve the quality of maternal care.

In our study, we found that primiparous women who gave birth in the sitting birth position had a shorter duration of the second stage of labour, which supported the results of the previous studies. Ganapathy et al. [12] found a similar reduction in the duration of the second stage of labour in the sitting position group compared to the lithotomy position group. In addition, a prior study conducted in China [30] assigned 112 primiparous women to the sitting or lithotomy position groups during the second stage of labour, and reported that the duration of the second stage of labour was reduced by an average of 20 min in the sitting group, which were in line with our findings. Several possible explanations for the shortened duration of the second stage of labour in the sitting position were proposed [31,32]. Firstly, the intensity of uterine contractions was stronger when women gave birth in a sitting position. Additionally, the sitting position could take advantage of gravity and facilitate the descent of the foetal head, thereby shortening the duration of the second stage of labour. A prolonged second stage of labour may lead to adverse maternal and neonatal outcomes and other delivery complications, and thus it was necessary to take effective interventions to shorten the duration of the second stage of labour [5]. Moreover, the available evidence also suggested the sitting position, which has the potential to promote labour progression and decrease adverse complications, as a preferable birth position option during the second stage of labour, especially for primiparous women [24,32].

In addition, our results suggested that the sitting position could promote spontaneous vaginal births and reduce episiotomies for primiparous women. Some studies also reported similar results, the findings indicated that women who adopted a sitting position were less likely to have instrumental birth and an episiotomy, and were more likely to have spontaneous vaginal births, which also led to lower perineal pain scores compared with women in the lithotomy position [12,17]. The findings from the current study indicated that the sitting position could potentially enhance the natural progression of labour, minimise unnecessary interventions, lead to improved maternal outcomes, and contribute to greater childbirth satisfaction among childbearing women. This also aligns with the ACOG committee’s recommendations of reducing unnecessary interventions during childbirth and promoting a more positive childbirth experience [33]. Therefore, it is recommended that the sitting position may be a favourable birth position option during the second stage of labour for women. Additionally, we did not find any significant difference among the primiparous women cohort and multiparous women cohort in terms of the perineal injuries, postpartum 2h-haemorrhage, Apgar scores at 5 min and 10 min, as well as the cord artery pH. It also indicated that the sitting position did not increase the risk of perineal injuries among multiparous women, which was consistent with the previous results [10]. In general, perineal injuries, as a common complication after vaginal birth, were associated with multiple negative maternal outcomes such as perineal pain, more blood loss during labour, pelvic floor injury, and urinary incontinence [22,34]. However, the existing studies did not differentiate the effects of maternal positions on these outcomes, thus, more related research is warranted.

For the primiparas in our study, we found that women in the sitting position cohort reported higher overall CEQ scores, which indicated a more positive childbirth experience in comparison to women in the lithotomy position. Further, based on the higher scores at all four dimensions of the CEQ for women who gave birth in the sitting position, the results showed that the sitting position could help women have more satisfactory professional support and gain better self-ability, self-perception, and sense of participation. The results were similar to a prior study that aimed to compare the effects of maternal birthing experience in the sitting versus the lithotomy position during the second stage of labour [12]. They found that women who adopted the sitting position reported a favourable birthing experience, as indicated by the lower intensity of labour pain measured by the Visual Analogue Pain Scale. Thus, the fact that the sitting position could reduce the women’s pain level may be a potential reason for a more positive childbirth experience. Another study also indicated that the freedom of movement and birth in upright positions during labour could increase the birth comfort of women and result in a better experience [16]. In recent years, positive childbirth experience has become an essential indicator for assessing high-quality maternal care in clinical practice [35]. A positive childbirth experience can enhance women’s satisfaction with labour, promote neonatal growth, and support postpartum recovery [36]. However, a negative childbirth experience can lead to postpartum depression, fear of subsequent childbirth, and other adverse maternal and neonatal outcomes [37,38]. Thus, it is necessary to adopt effective interventions to improve the women’s childbirth experience. Based on our findings, it is recommended that women assume the sitting position during the second stage of labour for a positive childbirth experience. Healthcare providers are also encouraged to consider the sitting position in the present management of maternal positions in clinical practice to improve the quality of maternal care.

### Strengths and Limitations

Our study not only examined the effects of the sitting position on maternal and neonatal outcomes but also assessed childbirth experience using a valid instrument, offering a comprehensive overview of the effects of the sitting position during the second stage of labour. Given this, our results could provide a reference for both healthcare providers and women to choose the appropriate birth position during the second stage of labour for a positive childbirth experience and better maternal and neonatal outcomes, especially for primiparous women. There are also several limitations in our study. First, our findings were derived from a single-centre study, which could potentially limit the generalisability of the results. Nevertheless, this study may facilitate improved planning for future multi-centre trials. Second, the sample of multiparous women is small, which may affect the reliability of the results. Third, our study enrolled low-risk and 20–35-year-old women based on the inclusion criteria, which also influenced the generalisability of the results among pregnant women. Thus, we suggest that future studies with larger samples could be conducted. In addition, the childbirth experience was evaluated by questionnaires, lacking in-depth exploration of mothers’ perceptions regarding sitting positions. Consequently, there is a need for additional qualitative studies or mixed-method studies in the future.

## 5. Conclusions

The results of our study indicated that primiparous women who gave birth in the sitting birth position in the second stage of labour had a shorter duration of the second stage of labour, higher rates of spontaneous vaginal births, fewer episiotomies, and a more positive childbirth experience compared with those who assumed the lithotomy position. In addition, there was not any significance among the maternal and neonatal outcomes for multiparous women to adopt the sitting position in the second stage of labour. Thus, our findings provided an important reference for women to choose the sitting position in the second stage of labour for a positive childbirth experience. Healthcare providers could also give advice to women based on our findings, which could also provide an innovative value for the clinical management of childbirth positions. We believe it will be also useful for healthcare providers to offer a high-quality maternal service for women.

## Figures and Tables

**Figure 1 healthcare-11-02996-f001:**
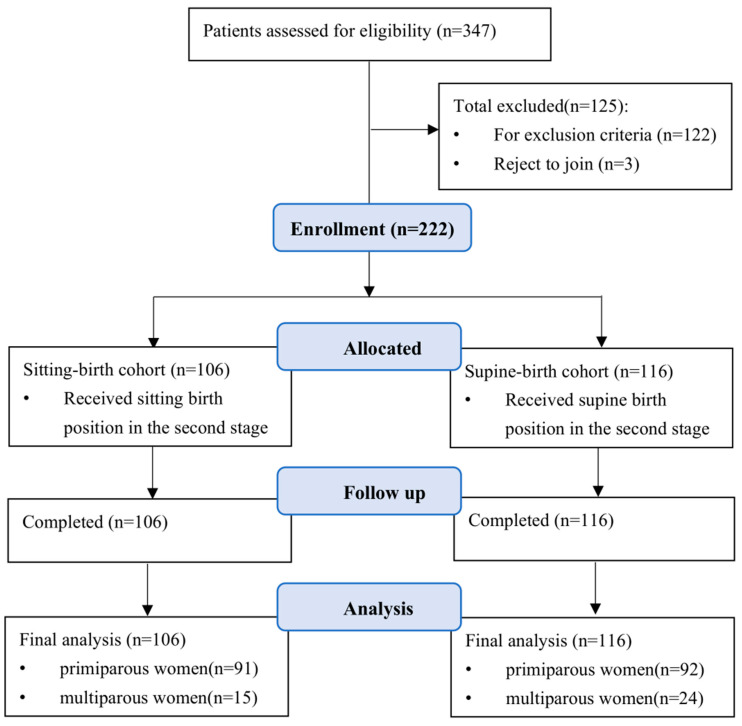
Flowchart of the sample selection procedure.

**Table 1 healthcare-11-02996-t001:** Demographic characteristics and birth status.

Variables	Total n (%) N = 222	Sitting-Birth Cohort n (%) N = 106	Supine-Birth Cohort n (%) N = 116
**Demographic characteristics**			
**Parity**	222	106	116
Primipara	183 (82.4)	91 (85.8)	92 (79.3)
Multipara	39 (17.6)	15 (14.2)	24 (20.7)
**Age (year), M ± SD**	30.97 ± 2.66	30.67 ± 2.55	39.75 ± 0.94
Primipara	30.48 ± 2.57	30.26 ± 2.46	30.68 ± 2.68
Multipara	33.31 ± 1.60	33.13 ± 1.55	33.42 ± 1.66
**Education**			
Primipara	183	91	92
High school or below	1 (0.5)	0 (0)	1 (1.1)
Bachelor’s or junior college	99 (54.1)	52 (57.1)	47 (51.1)
Master’s or doctorate	83 (45.4)	39 (42.9)	44 (47.8)
Multipara	39	15	24
High school or below	1 (2.6)	0 (0)	1 (4.2)
Bachelor’s or junior college	21 (53.8)	8 (53.3)	13 (54.2)
Master’s or doctorate	17 (43.6)	7 (46.7)	10 (41.7)
**Gestation weeks, M ± SD**	39^+5^ ± 0.97	39^+4^ ± 1.42	39^+5^ ± 0.94
Primipara	39^+5^ ± 0.98	39^+5^ ± 1.04	39^+6^ ± 0.92
Multipara	39^+3^ ± 0.89	39^+3^ ± 0.76	39^+3^ ± 0.97
**Caesarean section history**	2 (0.9)	0 (0)	2 (1.7)
Primipara	0 (0)	0 (0)	0 (0)
Multipara	2 (0.9)	0 (0)	2 (8.3)
**Body mass index (BMI) M ± SD (before pregnancy)**	21.51 ± 2.83	21.57 ± 2.99	21.46 ± 2.69
Primipara	21.50 ± 2.91	21.45 ± 3.09	21.55 ± 2.73
Multipara	21.58 ± 2.44	22.30 ± 2.18	21.13 ± 2.53
**Weight gain**	14.02 ± 7.58	13.32 ± 6.66	14.66 ± 8.31
Primipara	14.02 ± 7.09	13.13 ± 4.32	14.89 ± 8.98
Multipara	14.03 ± 9.65	14.47 ± 14.53	13.75 ± 5.01
**Position of foetus**			
Primipara	183	91	92
LOA	139 (76.0)	74 (81.3)	65 (70.7)
ROA	44 (24.0)	17 (18.7)	27 (29.3)
Multipara	39	15	24
LOA	35 (89.7)	13 (86.7)	22 (91.7)
ROA	4 (10.3)	2 (13.3)	2 (8.3)
**Birth status**			
**Oxytocin use**	94 (42.3)	44 (41.5)	50 (43.1)
Primipara	183	91	92
Yes	82 (44.8)	40 (44.0)	42 (45.7)
No	101 (55.2)	51 (56.0)	50 (54.3)
Multipara	39	15	24
Yes	12 (30.8)	4 (26.7)	8 (33.3)
No	27 (69.2)	11 (73.3)	16 (66.7)
**Epidural analgesia**	142 (64.0)	68 (64.2)	74 (63.8)
Primipara	183	91	92
Yes	127 (69.4)	60 (65.9)	67 (72.8)
No	56 (30.6)	31 (34.1)	25 (27.2)
Multipara	39	15	24
Yes	15 (38.5)	8 (53.3)	7 (29.2)
No	24 (61.5)	7 (46.7)	17 (70.8)
**Baby’s birth length (cm), M ± SD**	49.68 ± 1.56	49.42 ± 1.71	49.93 ± 1.36
Primipara	49.67 ± 1.38	49.53 ± 1.40	49.80 ± 1.36
Multipara	49.77 ± 2.23	48.73 ± 2.98	50.42 ± 1.28
**Baby’s birth weight (g), M ± SD**	3255.18 ± 352.23	3215.85 ± 360.68	3291.12 ± 341.94
Primipara	3220.82 ± 340.17	3187.58 ± 333.63	3253.70 ± 345.18
Multipara	3416.41 ± 367.35	3387.33 ± 472.62	3434.58 ± 293.42
**Position in the first stage of labour**			
Freestyle position	187 (84.6)	93 (87.7)	94 (81.7)
upright	1 (0.5)	1 (0.9)	0
Supine	33 (14.9)	12 (11.3)	21 (18.3)
Missing	1	0	1
**Complications (low-risk)**			
Yes	152 (68.5)	79 (74.5)	73 (62.9)
No	70 (31.5)	27 (25.5)	43 (37.1)

Note: LOA: left occiput-anterior; ROA: right occiput-anterior.

**Table 2 healthcare-11-02996-t002:** Comparison of primary and secondary physiologic outcomes of childbirth between the sitting-birth cohort and supine-birth cohort.

Variables	Primiparous Women					Multiparous Women				
	Sitting-Birth Cohort n (%) N = 91	Supine-Birth Cohort n (%) N = 92	χ^2^	t/Z	*p*	Sitting-Birth Cohort n (%) N = 15	Supine-Birth Cohort n (%) N = 24	χ^2^	t/Z	*p*
Duration of second stage (min) ME IQR	50.00 (47)	76.00 (61)		−3.657 ^b^	0.000	NS	NS			NS
Duration of first stage (min) ME IQR	395.00 (213)	370.00 (295)		−747 ^b^	0.455	NS	NS			NS
Duration of first and second stage (min) M ± SD	NS	NS			NS	246.67 ± 112.17	347.04 ± 216.88		−1.654 ^a^	0.107
Birth mode	N = 91	N = 92	11.623		0.001	N = 15	N = 24			NS
Spontaneously	85 (93.4)	69 (75.0)				15 (100)	24 (100)			
Vaginal midwifery	6 (6.6)	23 (25.0)				0	0			
CS	0 (0)	0 (0)								
Perineal injuries	N = 70	N = 49	1.320		0.725	N = 14	N = 23	2.939		0.230
Complete	10 (14.3)	5 (10.2)				2 (14.3)	9 (39.1)			
First degree	33 (47.1)	23 (46.9)				9 (64.3)	9 (39.1)			
Second degree	26 (37.1)	21 (42.9)				3 (21.4)	5 (21.7)			
Three/third degree	1 (1.4)	0 (0)				0(0)	0(0)			
Episiotomy	N = 91	N = 92	11.263		0.001	N = 15	N = 24	0.119		0.731
Yes	21 (23.1)	43 (46.7)				1 (6.7)	1 (4.2)			
No	70 (76.9)	49 (53.3)				14 (93.3)	23 (95.8)			
Postpartum 2h-haemorrhage			0.201		0.654			2.514		0.113
<500 mL	82 (90.1)	81 (88.0)				12 (80)	23 (95.8)			
≥500 mL	9 (9.9)	11 (12.0)				3 (20)	1 (4.2)			
Apgar										
<7 at 1, 5, 10 min	0 (0)	0 (0)				0 (0)	0 (0)			
Cord artery pH										
<7.00	0 (0)	0 (0)				0 (0)	0 (0)			
Missing	1	5								

Note: ^a^
*t*-test, ^b^ Mann–Whitney U test, NS: Not Suitable.

**Table 3 healthcare-11-02996-t003:** Comparison of maternal childbirth experience between the sitting-birth cohort and supine-birth cohort.

Variables	Primiparous Women				Multiparous Women			
	Sitting-Birth Cohort N = 91	Supine-Birth Cohort N = 92	t/Z	*p*	Sitting-Birth Cohort N = 15	Supine-Birth Cohort N = 24	t/Z	*p*
CEQ, M ± SD	3.26 ± 0.35	2.94 ± 0.44	5.421 ^a^	0.000	3.34 ± 0.42	3.09 ± 0.37	1.842 ^a^	0.074
Dimensions_1 Professional support	3.59 ± 0.44	3.28 ± 0.63	−3.337 ^b^	0.001	3.56 ± 0.46	3.48 ± 0.51	−0.333 ^b^	0.739
Dimensions_2 Self-ability	3.15 ± 0.45	2.81 ± 0.54	4.601 ^a^	0.000	3.33 ± 0.48	3.02 ± 0.47	1.953 ^a^	0.059
Dimensions_3 Self-perception	2.92 ± 0.47	2.54 ± 0.55	4.955 ^a^	0.000	3.14 ± 0.68	2.64 ± 0.38	2.581 ^a^	0.019
Dimensions_4 Sense of participation	3.28 ± 0.55	3.05 ± 0.49	−3.512 ^b^	0.000	3.19 ± 0.55	3.10 ± 0.58	0.492 ^a^	0.626

Note: ^a^
*t*-test, ^b^ Mann–Whitney U test.

**Table 4 healthcare-11-02996-t004:** Regression analysis of the sitting position on maternal and neonatal outcomes among primiparous women.

Variables	Primiparous Women			
	Lithotomy Position	B/B (95%CI)	Adjusted OR (95% CI)	*p*
Duration of second stage (min) ME IQR ^a^	Reference	19.271 (8.535, 30.007)	NS	0.001
Duration of first stage (min) ME IQR ^a^	Reference	36.768 (−22.268, 95.804)	NS	0.221
CEQ scores ^a^	Reference	−0.278 (−0.395, −0.160)	NS	0.000
Birth mode ^b^	Reference	−1.337	0.252 (0.092, 0.694)	0.008
Perineal injuries ^b^	Reference			
Complete	Reference	0.361	1.435 (0.426, 4.832)	0.560
First degree	Reference	0.089	1.093 (0.499, 2.397)	0.824
Second degree	Reference	−0.285	0.752 (0.343, 1.651)	0.478
Episiotomy ^b^	Reference	−1.029	0.357 (0.181, 0.706)	0.003
Postpartum 2h-haemorrhage ^b^	Reference	−0.110	0.896 (0.319, 2.518)	0.835

Note: Data adjusted for gestation weeks, age, BMI, weight gain, baby’s birth weight, oxytocin use, epidural analgesia, and complications; ^a^: multiple linear regression; ^b^: logistic regression; NS: not suitable.

## Data Availability

The data that support the findings of this study are available from the corresponding author.

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
