# Peer review of "Effects of Using Sitting Position versus Lithotomy Position during the Second Stage of Labour on Maternal and Neonatal Outcomes and the Childbirth Experience of Chinese Women: A Prospective Cohort Study"

_healthcare, 2023, doi:10.3390/healthcare11222996_

Round 1

Reviewer 1 Report

Comments and Suggestions for Authors

The authors performed a prospective study with the aim to explore whether the sitting position would influence the neonatal and maternal outcomes.

They found out that primary parous women who gave birth in sitting position had a shorter duration of the second stage of labour. This is in concordance with many other studies. Several studies and reviews suggest several possible benefits for upright posture in women without epidural anaesthesia, such as a very small reduction in the duration of second stage of labour (mainly from the primigravid group), reduction in episiotomy rates and assisted deliveries. However, there seems to be  an increased risk of higher blood loss and there may be an increased risk of second degree tears.  In view of the variable risk of bias in this trial, well-designed protocols are needed to ascertain the true benefits and risks of various birth positions.

The auhtors enrolled 222 pregnant women and allocated 106 in a sitting birth position and 116 in a supine birth position. The participants were divided into the sitting position, and the lithotomy position cohort to their own free will.

It is unbelievable that in a cohort of 222 deliveries, there is no forceps or vacuum assisted deliveries especially in the view that 70% of the women have epidural analgesia .Nearly half of them received oxytocin infusion. 

The use of fundal pressure during the second stage of labor is a cause of great concern - In this survey it isn't even mentioned.

The literature is not appropriate I was not able to find number 12 number17 number 7

Author Response

1.The authors performed a prospective study with the aim to explore whether the sitting position would influence the neonatal and maternal outcomes.

They found out that primary parous women who gave birth in sitting position had a shorter duration of the second stage of labour. This is in concordance with many other studies. Several studies and reviews suggest several possible benefits for upright posture in women without epidural anaesthesia, such as a very small reduction in the duration of second stage of labour (mainly from the primigravid group), reduction in episiotomy rates and assisted deliveries. However, there seems to be an increased risk of higher blood loss and there may be an increased risk of second-degree tears.  In view of the variable risk of bias in this trial, well-designed protocols are needed to ascertain the true benefits and risks of various birth positions.

 Response:

   Thanks for your insightful comment regarding the variable risk of bias in our trial. We appreciate your feedback and fully recognize the importance of well-designed protocols. The variable risk of bias in our study was considered through setting up the strict inclusion and exclusion criteria and conducting the multivariate linear regression and logistic regression to control possible confounders.

2.The authors enrolled 222 pregnant women and allocated 106 in a sitting birth position and 116 in a supine birth position. The participants were divided into the sitting position, and the lithotomy position cohort to their own free will.

It is unbelievable that in a cohort of 222 deliveries, there is no forceps or vacuum assisted deliveries especially in the view that 70% of the women have epidural analgesia. Nearly half of them received oxytocin infusion. 

 Response:

  Thanks for comments. The birth mode of “Vaginal midwifery” in the study refers to “forceps”, thus there is nearly 30 forceps deliveries among the participants.

3.The use of fundal pressure during the second stage of labor is a cause of great concern - In this survey it isn't even mentioned.

 Response:

  Thanks indeed. The fundal pressure was not used in our study. The application of manual fundal pressure to facilitate childbirth during the second stage of labor is not recommended based on the World Health Organization (WHO) recommendations for augmentation of labor because of the possible potential for harm to mother and baby.   

4.The literature is not appropriate I was not able to find number 12 number17 number 7. 

 Response:

  Thanks very much. We have added the web address of number 12 number17 number 7. And number 12 we have submitted the PDF.

Reviewer 2 Report

Comments and Suggestions for Authors

This is a cohort study comparing two different position during labour with regard to their overall effect on clinical outcomes. This is an interesting study. I only have a few comments to add:

1) Line 110-111: please specify in which hospital the study was conducted. If multicentre, state the number of the centres.

2)Lines 173-174: are these numbers correct? You only had 222 women, how can you have two arms of 206 and 216?

3)Line 214: p<0.001

Comments on the Quality of English Language

Language is fine

Author Response

1) Line 110-111: please specify in which hospital the study was conducted. If multicentre, state the number of the centres.

Response:

Thank you so much. We have revised it.

“This was a prospective cohort study conducted from February to June 2023 in Pe-king University People's Hospital, Beijing, China.”

2)Lines 173-174: are these numbers correct? You only had 222 women, how can you have two arms of 206 and 216?

Response:

Thank you and sorry for the error caused. We revised the numbers in this section.

“106 in the sitting position cohort and 116 in the lithotomy position cohort”

3)Line 214: p<0.001

Response:

Thank you so much. We have revised it.

Reviewer 3 Report

Comments and Suggestions for Authors

Dear authors,

It is a pleasure to greet you and express my sincere congratulations for your significant contribution to the field of healthcare through your article. Your dedication to this research is evident and serves as a testament to your commitment to improving healthcare.

I have had the honor of reviewing your article, and I have been impressed by its subject matter. Your approach and the results presented are valuable and have the potential to make a significant difference in the scientific community and healthcare at large.

As a reviewer, my goal is to help your work reach its full potential. Therefore, I have identified some suggestions and observations that I believe can further strengthen your research. These recommendations are intended to enhance the clarity, relevance, and contribution of your article.

I sincerely encourage you to consider these comments and address the issues mentioned in your review. I am confident that, with your expertise and dedication, you can make the necessary adjustments to make this article an even more valuable contribution to the field of healthcare.

I appreciate your hard work and dedication to research, and I look forward to your response and the revisions you will undertake. Together, we can continue to advance healthcare and knowledge in this area.

Thank you once again for your contribution.

Sincerely,

**Abstract**

1. Please, follow the publication guidelines and create a structured summary.

2. Use MesH terms as keywords.

**Introduction**

3. Rewrite the introduction. It is too dry to read due to overly lengthy paragraphs.

4. Sections from line 54 to 86, in which studies and results of other research are discussed, are better suited for the discussion section rather than the introduction. Rewrite the introduction considering these suggestions.

**Methodology**

5. Provide a more detailed description of the questionnaire used and how and when it was administered to the study participants.

6. What tools and/or tests did you use to check the normality of variables? Why did you combine parametric and non-parametric tests in your study?

**Results**

7. The data in the "The demographic and clinical characteristics of participants" section do not match the data presented in Figure 1. There is a calculation error. Please correct it.

8. For the variable "weeks of gestation," please format the data as "weeks+days."

9. Some sections of Figure 1 appear to be cut off. Please edit Figure 1 to ensure readability for the reader.

10. Add a table footnote in Table 1 clarifying the use of abbreviations.

11. Why are you analyzing the results based on parity? Analyze the questionnaire results in relation to both groups (giving birth in lithotomy and giving birth while standing).

**Discussion**

12. Please start the discussion by restating the research objectives.

13. The conclusions drawn in lines 285-295 of the discussion are not reflected in the results. Add a comparative table of overall results without dividing by parity in the results section, along with the statistical tests that led to these conclusions.

**Limitations**

14. In this section, include that the inclusion criteria were very strict, and participation in the study was limited to low-risk women, which means the results may not be applicable to pregnant women in general.

**Bibliography**

15. There are errors in the bibliography format. Please pay attention to the format and, for example, add bold when necessary.

Author Response

**Abstract**

1. Please, follow the publication guidelines and create a structured summary.

2. Use MesH terms as keywords.

Response:

Thank you very much. We have revised the structured summary in the manuscript according to the publication guidelines.

Thanks, we have used available Mesh terms as the keywords.

"Sitting position; labor stage, second; term birth; maternal and neonatal outcomes; childbirth experience; Prospective Studies"

**Introduction**

3. Rewrite the introduction. It is too dry to read due to overly lengthy paragraphs.

4. Sections from line 54 to 86, in which studies and results of other research are discussed, are better suited for the discussion section rather than the introduction. Rewrite the introduction considering these suggestions.

Response:

Appreciate your comments very much. We re-wrote it in the manuscript.

**Methodology**

5. Provide a more detailed description of the questionnaire used and how and when it was administered to the study participants.

6.What tools and/or tests did you use to check the normality of variables? Why did you combine parametric and non-parametric tests in your study?

Response:

Thank you indeed. We have provided the more detailed description of the questionnaire used and how and when it was administered to the study participants. 

The test of normality was used to check the normality of variables. For continuous variables, such as the duration of the second stage of labour, the test of normality was used to check the normality of variables, the student’s t-test (parametric tests) was utilized to compare the differences between groups if the data followed the normal distribution. The Mann-Whitne U test (non-parametric tests) were conducted if the data did not conform the normal distribution.

**Results**

7. The data in the "The demographic and clinical characteristics of participants" section do not match the data presented in Figure 1. There is a calculation error. Please correct it.

Response:

Thank you and sorry for the error caused. We corrected it.

8. For the variable "weeks of gestation," please format the data as "weeks+days."

Response:

Thank for your comments. We have descripted the variable "weeks of gestation" as "weeks+days."

9. Some sections of Figure 1 appear to be cut off. Please edit Figure 1 to ensure readability for the reader.

Response:

Sincere thanks. We revised it.

10. Add a table footnote in Table 1 clarifying the use of abbreviations.

Thank you so much. We have added the table footnote to clarify the use of abbreviations.

“Note: LOA: left occiput-anterior; ROA: right occiput-anterior.”

11. Why are you analyzing the results based on parity? Analyze the questionnaire results in relation to both groups (giving birth in lithotomy and giving birth while standing).

Response:

Thanks for raising this question. We conducted analysis by parity is mainly due to the clinical significance of birth management among women of different parity. The characteristics of labor in primiparous and multiparous women differ significantly, thus there is a significant heterogeneity in the primary and secondary outcomes between primiparous and multiparous women. Therefore, data analysis was conducted by parity.

**Discussion**

12. Please start the discussion by restating the research objectives.

Response:

Thanks for your comments. We revised it.

“This paper was designed to explore the effects of using sitting position versus lithotomy position during the second stage of labor on maternal and neonatal out-comes, as well as women’s childbirth experience, which aimed to provide the reference for the practice of childbirth positions and improve the quality of maternal care.”

13. The conclusions drawn in lines 285-295 of the discussion are not reflected in the results. Add a comparative table of overall results without dividing by parity in the results section, along with the statistical tests that led to these conclusions.

Response:

**Limitations**

14. In this section, include that the inclusion criteria were very strict, and participation in the study was limited to low-risk women, which means the results may not be applicable to pregnant women in general.

Response:

Thank you indeed. We added it to the limitation.

“Third, our study enrolled the low-risk and 20-35 years old women based on the inclusion criteria, which also influenced the generalization of the results among pregnant women.”

**Bibliography**

15. There are errors in the bibliography format. Please pay attention to the format and, for example, add bold when necessary.

Response:

Thank you so much. We revised it.

Reviewer 4 Report

Comments and Suggestions for Authors

The chosen topic is of interest to medical services, but the presentation needs to be improved.   

-       Line 22 - Wrong total women included in the study; 

-       Lines 27 and 33 – In the abstract, if not statistically significant, it should not be included; 

-       Incomplete keywords; 

-       Line 129 Please specify low-risk complications

-       Line 134: Why was the Apgar score chosen at 10 minutes?

-       Line 152 – Instead of second labor, insert the second stage of labor; 

-       Line 156-157 What did you mean by “Eventually, a total of 216 participants were needed”?; 

-       Lines 173-174 - Wrong number of participants; 

-       Line 198 In flowchart 1, complete the sentence “second stage of labor”; 

-       In line 223, you mentioned the Apgar score at 1 minute, but in the study variable, you don't mention it

-       Why is the Bishop score not mentioned in the study?

-       What was the number of newborns admitted to the NICU?

Kind regards

Author Response

1. Line 22 - Wrong total women included in the study.

Response:

Thank you and sorry for the error. We revised it.

2. Lines 27 and 33 – In the abstract, if not statistically significant, it should not be included.

Response:

Thank you so much. We revised it in the section.

3. Incomplete keywords.

Response:

Thank you indeed. We added the keywords in the manuscript.

“Sitting position; labor stage, second; term birth; maternal and neonatal outcomes; childbirth experience; Prospective Studies.”

4. Line 129 Please specify low-risk complications

Response:

Thank you so much. We have specified the low-risk complications including premature rupture of membrane, mild pre-eclampsia, pregnancy with diabetes, pregnancy with mild arrhythmia and pregnancy complicated with hysteromyoma.

5. Line 134: Why was the Apgar score chosen at 10 minutes?

Response:

Thanks for your comments very much. We chose the Apgar score at 1,5 and 10 minutes based on the “Guideline of normal birth” conducted by Chinese Medical Association. The guideline recommended recording the Apgar scores at 1, 5, and 10 min to assess the newborn health.

6. Line 152 – Instead of second labor, insert the second stage of labor.

Response:

Thanks indeed. We revised it.

7. Line 156-157 What did you mean by “Eventually, a total of 216 participants were needed”?

Response:

Thanks for your comments. We used the sample size formula for cohort studies and substituted the appropriate metrics to calculate a total sample size of 216 required.

8. Lines 173-174 - Wrong number of participants.

Response:

Thanks very much. We revised it.

“106 in the sitting position cohort and 116 in the lithotomy position cohort.”

9. Line 198 In flowchart 1, complete the sentence “second stage of labor”;

Response:

Thank you indeed. We completed the sentence “second stage of labor”.

10. In line 223, you mentioned the Apgar score at 1 minute, but in the study variable, you don't mention it.

Response:

Thanks very much. We have added it in the section.

“Newborn Apgar score at one, five and ten minutes postpartum,”

11.Why is the Bishop score not mentioned in the study?

Response:

Thanks for your comments. All childbearing women in the study were assessed by the obstetricians through the Bishop scores and the enrolled participants were eligible for vaginal trial of labor, as noted in the inclusion criteria “intended to have a spontaneous vaginal birth” in part 2.1.

12. What was the number of newborns admitted to the NICU?

Response:

Thanks for your comments. There was no newborn admitted to the NICU.

Round 2

Reviewer 3 Report

Comments and Suggestions for Authors

Thank you for your paper.

Reviewer 4 Report

Comments and Suggestions for Authors

The article has been modified as recommended and is eligible for publication.

Kind regards